# Biomaterial-Based Therapeutic Strategies for Obesity and Its Comorbidities

**DOI:** 10.3390/pharmaceutics14071445

**Published:** 2022-07-11

**Authors:** Jing Li, Hongli Duan, Yan Liu, Lu Wang, Xing Zhou

**Affiliations:** 1School of Pharmacy and Bioengineering, Chongqing University of Technology, Chongqing 400054, China; ljing@stu.cqut.edu.cn (J.L.); julyduan@stu.cqut.edu.cn (H.D.); cqutliuyan@163.com (Y.L.); 2Institute of Materia Medica and Center of Translational Medicine, College of Pharmacy, Army Medical University, Chongqing 400038, China; 3Chongqing Key Laboratory of Medicinal Chemistry & Molecular Pharmacology, Chongqing University of Technology, Chongqing 400054, China

**Keywords:** biomaterial, drug delivery, obesity

## Abstract

Obesity is a global public health issue that results in many health complications or comorbidities, including type 2 diabetes mellitus, cardiovascular disease, and fatty liver. Pharmacotherapy alone or combined with either lifestyle alteration or surgery represents the main modality to combat obesity and its complications. However, most anti-obesity drugs are limited by their bioavailability, target specificity, and potential toxic effects. Only a handful of drugs, including orlistat, liraglutide, and semaglutide, are currently approved for clinical obesity treatment. Thus, there is an urgent need for alternative treatment strategies. Based on the new revelation of the pathogenesis of obesity and the efforts toward the multi-disciplinary integration of materials, chemistry, biotechnology, and pharmacy, some emerging obesity treatment strategies are gradually entering the field of preclinical and clinical research. Herein, by analyzing the current situation and challenges of various new obesity treatment strategies such as small-molecule drugs, natural drugs, and biotechnology drugs, the advanced functions and prospects of biomaterials in obesity-targeted delivery, as well as their biological activities and applications in obesity treatment, are systematically summarized. Finally, based on the systematic analysis of biomaterial-based obesity therapeutic strategies, the future prospects and challenges in this field are proposed.

## 1. Introduction

Obesity is a chronic metabolic disease defined as an excessive or abnormal accumulation of body fat that adversely affects health. The incidence of obesity has continued to rise worldwide in the last few decades, causing a serious public health crisis [1,2]. The body mass index (BMI) is commonly used clinically to assess the degree of obesity, which is calculated by the formula: BMI = weight(kg)/height(m)^2^. A BMI above 25 kg/m^2^ is defined as overweight, above 30 kg/m^2^ is defined as obese, and over 35 kg/m^2^ is defined as severely obese [3]. According to relevant statistics, more than 1.5 billion adults worldwide are already overweight, of which at least 300 million are considered clinically obese. In the United States, nearly 40% of adults suffer from obesity, and the rate is as high as 18.5% in children and adolescents [4]. As a developing country, in China, with its rapid economic rise and the improvement of living standards, the consequent problem of excess nutritional intake has led to a rapid increase in obesity rates, with 11.1% of the population affected by obesity and 7.9% being teenagers [5,6]. Experts predict that, according to the current epidemic trend, by 2025, the obesity rate of the world’s population will exceed 39%, and the rate of severe obesity will exceed 15% [7].

Obesity can cause various short-term adverse effects on physiological functions, such as hypercholesterolemia, high triglycerides, insulin resistance, and increased peripheral vascular resistance. More seriously, by tracking large numbers of obese children and adolescents to adulthood, it has been found that obesity can lead to various complications, such as type 2 diabetes (T2D), hypertension, cardiovascular disease, and so on [8,9,10,11]. In addition, obesity also increases the incidence of tumors such as prostate, uterine, and breast cancers [12]. Based on etiology and pathogenesis, obesity can be simply divided into simple obesity and pathological obesity [13]. Simple obesity is linked to poor lifestyle behaviors. This group of people, on the one hand, overeat high-fat and high-calorie food; on the other hand, they live a sedentary life and lack physical exercise, thus leading to systemic fat accumulation. Pathological obesity is mainly caused by endocrine diseases, congenital diseases, metabolic syndrome, and others, including Cushing’s syndrome, hypothyroidism, hepatitis, etc. [14,15]. Some drugs, such as antipsychotic drugs, antidepressants, antiepileptic drugs, hormones, and other endocrine drugs, can also lead to pathological obesity. Some medicines, such as antipsychotic drugs, antidepressants, antiepileptic drugs, and hormones, can also lead to pathological obesity [16,17]. In fact, simple obesity can further aggravate and produce pathological changes if left untreated. It is evident that obesity and its related comorbidities not only seriously damage the physical and mental health of patients but also negatively affect economic and social development.

Bariatric surgery has been practiced for over 60 years. It has grown in popularity with the improvement and prevalence of laparoscopic surgery, for example, gastric bypass surgery, and implantable devices such as the adjustable Lap-band^®^ and the Realize Gastric band^®^ [18,19]. The procedures involved are usually highly invasive. They are mainly used to treat patients with severe obesity and one or more comorbidities. Although surgical treatment has improved the quality of survival and reduced the incidence of obesity-related deaths to some extent, it cannot be ignored that this induces long-term side effects, such as micronutrient deficiencies of iron, vitamin B12, folic acid, and vitamin D, as well as the development of associated disorders, such as anemia, neurological syndromes, dementia, and depression [20,21,22]. Behavioral intervention is a relatively safe way to lose weight. It requires patients to follow a scientific and appropriate diet to reduce energy intake and increase effective exercise intensity at the same time to consume excess body fat and energy. However, most of them are challenging to stick with for long. For those who are non-responsive to lifestyle intervention within six months and suffer from obesity-related diseases, pharmacological treatment is recommended [23,24].

## 2. Pharmacological Treatment of Obesity

There are three main mechanisms of action of anti-obesity drugs: one is by acting on neural pathways to suppress appetite, another is by acting on the gastrointestinal tract to inactivate lipase, thereby inhibiting fat absorption, and the third is by converting white adipose tissue (WAT) into brown adipose tissue (BAT) to increase energy consumption (Figure 1) [25,26]. Despite substantial financial investment and considerable effort, the development of anti-obesity therapeutic agents has not gone very well. The United States Food and Drug Administration (FDA) has approved a variety of anti-obesity drugs, such as aminorex, rimonabant, sibutramine, dexfenfluramine, etc. Unfortunately, these drugs have been recalled due to their low efficacy, high price, and various side effects, such as myocardial infarction, stroke, and severe neuropsychiatric side effects [27,28,29]. There are only a few drugs currently on the market approved for obesity treatment, including orlistat (Xenical^®^), phentermine (Adipex-P^®^), lorcaserin (Belviq^®^), liraglutide (Saxenda^®^), semaglutide (Wegovy^®^), naltrexone/bupropion sustained-release (Contrave^®^), and phentermine/topiramate extended-release (Qsymia^®^) (Table 1). For safety reasons, only three of them, orlistat, liraglutide, and naltrexone/bupropion (Mysimba^®^), are approved for use by the European Medicines Agency (EMA). Most of these available drugs focus on molecular targets in the central nervous pathway to reduce appetite, except for orlistat, which works on the gastrointestinal tract and inhibits fat absorption by suppressing lipase activity [30,31]. Orlistat and liraglutide are approved for long-term use, while others are used for short-term weight loss owing to their side effects [26,32]. Nevertheless, the usage of orlistat is limited to no more than two years due to adverse effects such as irregular bowel movements [33,34]. For drugs that act on the nervous system such as Phentermine, Lorcaserin, and Nal-trexone, they are mainly plagued by adverse effects such as insomnia, dizziness, nausea, and dry mouth [35,36,37,38,39,40]. Liraglutide is a glucagon-like peptide-1 receptor (GLP-1R) agonist that delays gastric emptying by enhancing insulin secretion and inhibiting glucagon secretion while reducing food intake through central appetite control. In addition to gastrointestinal side effects, liraglutide has been reported to increase the risk of pancreatitis as well as to increase heart rate [41,42,43]. Another GLP-1R agonist, semaglutide, was approved for marketing as an anti-obesity drug in recent years. It demonstrated outstanding weight loss benefits while significantly reducing classical risk factors, such as lipid and glucose levels and blood pressure for heart disease and diabetes, resulting in improved overall quality of life for patients. It was noted that the side effects of the drug were only mild to moderate nausea and diarrhea [44,45,46]. These two GLP-1R agonists are currently administered by injection, and oral formulations are now under development.

As the central pathways and pharmacological targets that regulate energy homeostasis are examined, including central pathways (e.g., the leptin–melanocortin axis and the opioid system), as well as specific signaling pathways of peripheral ligands acting on the central nervous system (e.g., the FGF21/FGFR1c/b-Klotho axis), several agents focusing on them are being investigated [47]. Leptin is a peptide hormone secreted mainly by WAT. It acts on the leptin receptor (LepRb) expressed in the hypothalamus to suppress appetite, increase energy expenditure, and inhibit adipogenesis. Protein tyrosine phosphatase 1B (PTP1B) is the key intracellular factor that negatively controls leptin signaling. A small-molecule inhibitor of PTP1B (an antisense drug called ISIS-PTP1BRX) has been shown to reduce body weight and glycated hemoglobin levels in subjects with T2D in clinical trials [48,49]. The central melanocortin system consists of neurons in the arcuate nucleus (ARC) that express pro-opiomelanocortin (POMC) and agouti-related protein (AgRP). The anorectic peptides α- and β-melanocyte-stimulating hormones (α-MSH and β-MSH) and the orexin AgRP secreted by these neurons target downstream neurons expressing the melanocortin-4 receptor (MC4R) to regulate physiological functions such as glucose homeostasis and thermogenesis. Genetic defects in the MC4R signaling pathway can lead to severe obesity [50,51]. The results from clinical experiments demonstrate that, in contrast to the previously developed alpha-MSH and the first-generation agonist MSH LY2112688, the second-generation MC4R agonist setmelanotide (also known as RM-493 or BIM-22493) has the ability to activate nuclear factor of activated T cells and restore this signaling pathway through selected MC4R variants [52]. Thus, it can effectively control overeating induced by genetic defects in the MC4R signaling pathway without significant side effects or adverse cardiovascular events, bringing hope to patients with central genetic obesity. Recent genetic evidence indicates that a locus in the mu-opioid receptor (MOR) gene OPRM1 is associated with dietary intake of fat [53], and observations from preclinical trials in several animal models have shown that the MOR antagonist naltrexone in combination with bupropion at a fixed dose exerts a synergistic weight loss effect, and GSK1521498 alone also has a weight reduction effect. However, unfortunately, these do not work well on human beings [54,55]. Fibroblast growth factor 21 (FGF-21) is a polypeptide generated mainly by the liver and adipose tissue, with pharmacological effects primarily in the regulation of glucolipid metabolism. In diet-induced obese (DIO) mice, intracerebroventricular administration of FGF21 increased the metabolic rate and insulin sensitivity [56]. FGF21 and its modified analogs were shown to significantly reduce body weight in both rodent and primate models [57,58,59]. In tests on primates, an FGF21 recombinant protein derivative named LY2405319 significantly induced weight loss by reducing food intake [60,61]. Obese T2D subjects receiving LY2405319 were found to have elevated levels of β-hydroxybutyrate, suggesting enhanced fatty acid oxidation and increased total energy expenditure, resulting in a slight decrease in body weight [62]. Despite the potent pharmacological effects of FGF-21, some individuals are expected to have FGF-21 resistance, so it would be beneficial to develop more effective FGF-21 agonists. Ongoing clinical trials and advanced basic research will provide a sound basis for developing FGF-21 agonists that have therapeutic value in metabolic diseases.

Given the diverse side effects of synthetic drugs, some researchers have focused on relatively less toxic phytochemicals, including resveratrol, curcumin, quercetin, capsaicin, and epigallocatechin gallate, which have shown some potential to combat obesity and associated comorbidities [63,64,65]. These substances decrease body weight mainly by reducing adipocyte formation, browning adipocytes, increasing lipolysis and energy expenditure, and inhibiting inflammation and oxidative stress. However, numerous problems, including low water solubility, poor stability, low bioavailability, and rapid enzyme metabolism in the gastrointestinal tract, liver, kidney, and other tissues, have limited their clinical use. Obese and overweight populations continue to rise worldwide, causing severe global health and economic burdens. Despite the increasing attention paid to them in medical research, there are still no highly effective means to prevent the development of these diseases. Therefore, there is an urgent need to explore safer and more efficient treatment strategies.

## 3. Biomaterial-Assisted Anti-Obesity Therapy

The evolution of materials science has laid the foundation for the development of biomedical materials. Drug delivery platforms prepared from a variety of materials, such as lipids, polymers, metals, and nonmetallic inorganic compounds (silica, graphene, etc.) [66,67,68], are being systematically used for therapeutic research on various diseases, including cardiovascular diseases [69], neurodegenerative diseases [70], cancer [71], autoimmunity, and so on [72,73]. These systems can not only deliver different types of drugs, such as chemicals, proteins, and nucleic acids, in a controlled manner but also promote the effective accumulation of drugs at the target site, thereby improving therapeutic efficacy and reducing toxic side effects.

This review mainly focuses on applying biomaterials in the treatment of obesity and its comorbidities. The main therapeutic strategies involved are as follows (Figure 2): (1) combating obesity by the inherent properties of certain materials, e.g., chitosan and its derivatives [74,75,76]; (2) covalently coupling small-molecule agents to natural or synthetic polymers, which can regulate the release rate and increase the drug’s stability [30]; and (3) delivering drugs by physically loading them into nanoparticles, hydrogels, transdermal microneedles, etc. Nanoparticles are commonly used as oral delivery vehicles for anti-obesity drugs, which can prevent drug inactivation in the gastrointestinal tract, improve drug stability, and improve the bioavailability of drugs by oral absorption [77,78,79,80,81]. Modification with antibodies, glycosyl groups, etc., can also endow the carriers with active targeting capabilities. Liposomes are one of the broadly studied nanocarriers, with a vesicular structure consisting of a lipid bilayer of phospholipids and cholesterol, and generally encapsulate hydrophilic drugs in the hollow cavity and hydrophobic drugs in a bimolecular membrane. For systemically administered drug-loaded liposomes, the surface decoration of the hydrophilic polymer polyethylene glycol (PEG) can inhibit their uptake by the reticuloendothelial system, reduce the renal clearance rate, and prolong the retention time of the drug in the systemic circulation [79]. Microneedles are needle-like structures with diameters ranging from tens of microns to a few millimeters. The application of microneedle technology or a microneedle patch enhances the skin penetration of therapeutic drugs in a minimally invasive way. It penetrates the stratum corneum but does not touch the dermis and nerve terminals, allowing the painless delivery of drugs to adipose tissue and inducing adipose tissue browning [82]. Compared to conventional delivery approaches, microneedle administration can improve patient compliance and the safety and efficacy of drug delivery [83,84]. This paper intends to review studies from the perspective of phytochemicals and synthetic and biological anti-obesity drugs in terms of their improved pharmacokinetics and their use in the treatment of obesity and its comorbidities with the aid of functional materials, expecting to provide a reference for the development of anti-obesity drugs with better efficacy and fewer side effects.

### 3.1. Biomaterials with Inherent Anti-Obesity Activity

Some biomaterials exert the effect of weight loss on their own without any drug loading, as listed in Table 2. Chitosan, a natural polysaccharide composed of glucosamine and N-acetylglucosamine copolymers, has been shown to have anti-obesity effects [85,86] and thus can be used as a lipid-lowering dietary supplement [87,88]. The anti-obesity effect of chitosan was previously thought to arise from its unique fat-binding properties, which interfere with the absorption of dietary lipids at the intestinal level [89,90]. However, recent studies suggest that the anti-obesity function of chitosan involves a more complex endocrine mechanism; that is, it works by regulating the concentrations of adipokines, including serum leptin and c-reactive protein (CRP) [73]. As mentioned above, leptin is a hormone secreted by adipocytes. When body fat increases or the body is in a high-energy state, the serum leptin level will increase, and the brain’s hypothalamus will receive a signal to stop eating. Leptin resistance refers to the fact that although the body produces a large amount of leptin, the leptin receptors in the brain are insensitive to leptin and fail to sense warning signals sent by adipocytes [91]. Chitosan oligosaccharide (COS) is a small-molecule derivative of chitosan with better water solubility than chitosan, and its absorption rate in the intestinal tract is close to 100% [92]. COS has varying biological activities, such as anti-inflammatory, anti-tumor, and antioxidant effects [93,94,95], and in several studies on obese animals, COS also showed valid hypolipidemic and anti-obesity effects [96,97,98]. A mechanistic study revealed that chitosan oligosaccharide capsules (COSCs) could alleviate the leptin resistance status, inhibit adipogenesis, and reduce lipid accumulation by activating the leptin signaling pathway (Janus kinase-2-signal transducer and activators of transcription-3, JAK2-STAT3). It is suggested that COSCs could be a potential candidate for obesity prevention or treatment [75]. Peroxisome proliferator-activated receptor gamma (PPARγ) is an important member of the nuclear receptor transcription factor superfamily, which is involved in the control of metabolic disorders (including obesity, insulin resistance, and cardiovascular disease) and plays a key role in regulating lipid metabolism [99,100]. It has been shown that COS can regulate the disorder of hepatic glucose and lipid metabolism by inhibiting obesity-related inflammatory responses and upregulating the expression of PPARγ. It reveals the potential application of COS in the prevention and treatment of glucolipid metabolism-related diseases [76].

Poly(lactide-co-glycolide) is a synthetic degradable polymer that can be processed into various forms, such as nanoparticles, hydrogels, and scaffolds, for drug delivery and tissue engineering [101,102,103,104]. PLG materials have good biocompatibility, and it was found that the porous PLG scaffold was well integrated with the tissue after implantation into the epididymal fat pad [105]. Surprisingly, implantation of the empty scaffold without loading any drug induced the increased expression of the proteins glucose transporter protein 1 and insulin-like growth factor I, involved in wound healing, as well as the increased expression of glucose transporter proteins Glut1 and Glut4, which regulate blood glucose levels, which in turn led to enhanced glucose uptake in the epididymal fat pad and reduced blood glucose levels in mice, confirming that the PLG scaffold could protect mice from diet-induced obesity and glucose intolerance [106]. It is evident that PLG scaffolds are a promising platform for treating advanced metabolic diseases, and their therapeutic function can be further enhanced by employing this material for pharmacological or cellular delivery.

In addition to polymer materials, some inorganic or metallic materials also have anti-obesity effects. Gold nanoparticles (AuNPs) have the advantages of easy surface modification, good stability, and non-cytotoxicity. In a study on high-fat-fed mice, compared with the untreated control group, intraperitoneal injection of AuNPs could significantly reduce the accumulation of abdominal fat and alleviate hyperlipidemia as well as poor glucose tolerance [107]. The underlying mechanism was related to its ability to decrease macrophage recruitment and activity in adipose tissue and liver and reduce the secretion of pro-inflammatory cytokines and tumor necrosis factor (TNF)-α, which are associated with obesity comorbidities. Further studies revealed a similar benefit of AuNP treatment in mice with pre-existing obesity, implying the promising potential of AuNPs in the treatment of obesity and obesity-induced glucolipid metabolism disorder [108]. Smilax glabra is a traditional Chinese medicine with various pharmacological properties, such as anti-diabetic, anti-jaundice, and anti-cancer activities. It was confirmed that gold nanoparticles synthesized with Smilax glabra rhizome had significant efficacy in obese diabetes rats induced by a high-fat diet and streptozotocin, helping restore damage to hepatocytes and cardiac veins and controlling weight gain [109]. Gold nanoshells have photothermal conversion capacity, which can convert the absorbed near-infrared (NIR) light into thermal energy, thus being used for photothermal lipolysis. For example, a polypyrrole-coated hollow gold nanoshell (HAuNS@PPy) was synthesized, where the polypyrrole was employed to improve the biocompatibility and the photothermal conversion efficiency of the material. The resultant HAuNS@PPy exhibited favorable photothermal stability and could effectively induce the thermo-mediated death of adipocytes [110]. In another study, hyaluronic acid was applied to modify the nanogold shell to improve its stability and biosafety, and adipocyte-targeting peptide (ATP) was decorated on the surface of the particles, which specifically binds to prohibitin located on the surface of adipocytes [111]. The obtained (HA-HAuNS-ATP) can target subcutaneous adipocytes after transdermal delivery and the ablate adipose tissue of C57BL/6 obese mice with NIR laser irradiation. In addition, porous colloids such as smectite clays and mesoporous silica also have unique bioactivities in regulating lipid metabolism. They show a weight loss effect in rodent obesity models by limiting the digestion of lipids through the disruption of the process by which digestive enzymes such as gastric and pancreatic lipases adhere to the surface of lipid droplets while also adsorbing dietary lipids and carbohydrates in the gastrointestinal tract to promote their excretion [112]. Nevertheless, further mechanistic studies are needed to facilitate the clinical application of these materials in preventing and treating obesity.

**Table 2 pharmaceutics-14-01445-t002:** Biomaterials with inherent anti-obesity activity.

Materials	Mechanism of Action	Characteristics	Main Outcomes	Reference
Chitosan and chitosan oligosaccharide	Upregulates the expression of serum leptin and CRP to inhibit adipogenesis and activates PPARγ expression to ameliorate glucose and lipid metabolism disorders	Biodegradable and low toxicity, with easily modifiable amino groups	Attenuates obesity and modulates glucose and lipid metabolism	[74,75,76]
PLG implants	Increases the expression of glucose transporter 1 and insulin-like growth factor 1 and increases glucose uptake	Synthetic, degradable, and good biocompatibility	Attenuates obesity and alleviates glucose intolerance	[106]
Au NPs	Reduces inflammation, regulates lipid metabolism, and ablates fatty tissue with near-infrared light	Cell regulation, good biocompatibility, and photothermal conversion capacity	Attenuates obesity and alleviates glucose intolerance	[107,108,109,110,111]
Smectite clays and mesoporous silica	Adsorbs digestive enzymes to limit lipid digestion and adsorbs fats and carbohydrates to promote their excretion	Porous colloidal structure	Attenuates obesity	[112]

### 3.2. Biomaterial-Encapsulated Phytochemicals for Anti-Obesity Treatment

A wide range of natural products can be valuable sources for developing anti-obesity drugs, which are characterized by diverse structures, relatively high activity, and mild side effects, among which phenolic acid, flavonoids, terpenoids, alkaloids, and other natural products have notable anti-obesity potential (Table 3). They inhibit adipose tissue formation, increase adipose tissue thermogenesis, and induce WAT browning [113,114]. However, their clinical application is hampered by unfavorable factors such as poor solubility, stability, and bioavailability [115].

#### 3.2.1. Resveratrol

Resveratrol is a natural polyphenolic compound with weight loss properties and is a representative drug for botanical anti-obesity treatment. However, the bioavailability of resveratrol is relatively low, with only 1–8% of free resveratrol remaining in the serum after intragastric administration in C57BL/6J mice [116]. This drug combats obesity mainly by inhibiting the growth of adipocytes, promoting their apoptosis, and promoting lipolysis. Y. J. Zu et al. prepared trans-resveratrol (R)-loaded, adipose stromal cell (ASC)-targeting peptide (CSWKYWFGECASC)-modified nanoparticles (L-Rnano) to induce the differentiation of ASCs into beige adipocytes [117]. Nanoencapsulation and targeted delivery significantly improved drug solubility, circulation time, and specific uptake by ASC. The animal experiments showed that L-Rnano was four times more efficient than unmodified control Rnano in targeting ASC in inguinal WAT of C57BL/6 mice while maintaining minimal liver accumulation and low hepatotoxicity. After five weeks, it was observed that the targeting group significantly induced ASC to differentiate into beige adipocyte and resulted in a 40% reduction in adiposity while improving glucose homeostasis and reducing the inflammatory response. In another study by the group, R-encapsulated lipid nanocarriers (R-nano) were compared with R-encapsulated liposomes (R-lipo), both of which are biocompatible and biodegradable [118]. The results showed that nanoencapsulation increased the uptake of R by 3T3-L1 adipose precursor cells, resulting in high expression of the browning marker uncoupling protein 1 (UCP1) and CD137 and low expression of the white marker IGFBP3. In terms of biological activity, R-lipo outperforms R-nano, probably due to its higher physical and chemical stability at room and body temperatures. In addition, nanoencapsulated resveratrol was prepared with starch particles of horse chestnut, water chestnut, and lotus stem, and the capsules were added to wheat flour to prepare snacks through an extrusion process [119]. The encapsulation process prevented the thermal degradation of resveratrol during preparation compared to the control group with free resveratrol added. Meanwhile, functional snacks containing encapsulated resveratrol showed significantly better antioxidant, anti-diabetic, and anti-obesity properties than the resveratrol-free ones.

#### 3.2.2. Capsaicin

Capsaicin is a natural active substance present in chili peppers, which has various biological effects, such as antioxidant, anti-inflammatory, anti-cancer, and hypolipidemic activities. It has been demonstrated in animal models and clinical studies that capsaicin has a beneficial impact on obesity and insulin resistance [120,121]. However, capsaicin in standard doses is highly irritating to the gastrointestinal tract and cannot be administered orally for long-term use. I. Lacatusu et al. prepared nanostructured lipid carriers (NLCs) to deliver capsaicin. The NLC was composed of flaxseed oil, oleoyl ethanolamine (OEA), and phenylalanine oleamide (PAO) [122]. Flaxseed oil has anti-inflammatory and triglyceride-lowering properties. OEA is an endogenous lipid that regulates feeding and body weight in vertebrates. PAO is a peroxisome receptor analog with weight loss properties. The system incorporates natural active compounds and endogenous lipids and is an anti-obesity formulation with better safety and tolerance. C. Bao et al. developed capsaicin-loaded α-lactalbumin (α-lac) micelles, named M(Cap), and used a microneedle patch for transdermal delivery that could melt at body temperature [123]. The micelles preferentially release capsaicin at the acidic pH of adipose tissue. Studies on DIO mice showed that the microneedle patch effectively delivered M(Cap) to the abdominal subcutaneous adipose tissue. The micelles were endocytosed by white adipocytes, resulting in significant weight loss. The mechanism of action is related to the activation of energy metabolism, increased mitochondrial biogenesis, and the induction of adipocyte browning.

**Table 3 pharmaceutics-14-01445-t003:** Biomaterial-encapsulated phytochemicals for anti-obesity treatment.

Drugs	Materials	Mechanism of Action	Characteristics	Main Outcomes	Reference
Resveratrol	Lipid nanoparticles modified with ASC-targeting peptides; nanocapsules prepared from starch particles	Induces browning of white adipocytes	Increases drug bioavailability and decreases toxicity	Attenuates obesity and reduces inflammatory response	[117,118,119]
Capsaicin	Liposomes; microneedle patches	Induces browning of white adipocytes and increases mitochondrial biogenesis to activate energy metabolism	Reduces drug irritation to the gastrointestinal tract and increases drug enrichment in local adipose tissue	Attenuates obesity and reduces inflammatory response	[122,123]
Caffeine	Microneedle patches	Reduces the levels of triglyceride, total cholesterol, and low-density lipoprotein and stimulates lipolysis	Avoids gastrointestinal absorption of the drug	Attenuates obesity	[124]
Allicin	DNA nanoflowers modified with adipo-8 aptamer	Induces browning of white adipocytes	Enhance the biological activity and stability of the drug	Attenuates obesity	[125]

#### 3.2.3. Caffeine and Allicin

Caffeine is a natural component of tea and coffee with anti-obesity effects and no adverse effects. However, the first-pass effect of caffeine after oral administration results in the low bioavailability of the drug. Transdermal administration has the benefit of bypassing the first-pass metabolism of the liver. Typically, biofilms are used to achieve transdermal administration of caffeine. However, caffeine undergoes a transition from the anhydrous to the crystalline form in biofilms, resulting in a maximum loading of no more than 5.5%. Using hyaluronic acid (HA) as a crystal growth inhibitor in combination with soluble microneedles inhibited the crystal growth of caffeine and allowed efficient loading of the drug [124]. After six weeks of administration to DIO mice, there were significant reductions in triglyceride, total cholesterol, and low-density lipoprotein (HDL) levels and substantial weight loss of approximately 12.8 ± 0.75%. To achieve adipocyte-targeted delivery of allicin, which is a phytochemical capable of inducing browning in adipose tissue, adipo-8 aptamer-modified DNA-nanoflower-allicin framework (NFA) was assembled using the isothermal rolling circle technique, which considerably enhanced the bioactivity and stability of allicin (Figure 3) [125]. Mechanistic studies have shown that the target of allicin is the G-quadruplex (G4) in the mitochondrial uncoupling protein-1 (UCP1) promoter, and adipo-8 and allicin play a synergistic role in the activation of targeted thermogenic genes. NFA’s subcutaneous injection can effectively promote adipocyte browning and systemic energy expenditure with minimal side effects.

The applications of biomaterials in the delivery of anti-obesity phytochemicals such as resveratrol, capsaicin, caffeine, and allicin are reviewed above. Compared to natural drugs, the preparation of resveratrol nanoparticles or lipid capsaicin overcomes the drug’s gastrointestinal irritation and improves its bioavailability. Additionally, loading caffeine into microneedles avoids the hepatic first-pass effect. Thus, phytochemical-loaded biocarriers have promising potential for obesity treatment.

### 3.3. Biomaterial-Encapsulated Synthetic Drugs for Anti-Obesity Treatment

This section reviews four synthetic anti-obesity drugs, namely, orlistat, rosiglitazone, thiopental sodium, and bindarit (Table 4). These drugs exert their anti-obesity effects mainly by inhibiting the activity of gastrointestinal lipase, inducing WAT browning, preventing lipid peroxidation, and reducing triglycerides and glucose levels in the body.

#### 3.3.1. Rosiglitazone

Rosiglitazone (Rosi) is a thiazolidinedione (TZD) anti-diabetic drug, which also activates PPARγ to induce WAT browning. However, the lack of specificity has caused toxic side effects on the liver, kidney, and brain, limiting its use in anti-obesity therapy [126,127]. Rosi, which is poorly water-soluble, was loaded into nanoparticles with a hydrophilic polyvinyl alcohol (PVA) surface layer and a poly(lactic-co-glycolic acid) (PLGA) core. With diameters of ~200 nm, these drug-loaded particles resided in liver Kupffer cells and macrophages in WAT after preferential uptake by circulating monocytes after systemic injection [128]. The drug is released specifically in acidified macrophage phagosomes, upregulates the expression of PPARγ target genes, and attenuates the obesity-induced inflammatory response in macrophages. Moreover, the agent does not alter the expression of genes related to lipid metabolism or cardiac function, indicating reduced side effects. As a monocyte- and macrophage-targeted PPARγ agonist delivery system, this platform provides a novel approach for treating macrophage-mediated inflammatory states associated with obesity, atherosclerosis, and other chronic diseases.

Photodynamic therapy (PDT) is a therapeutic method used to treat tumors as well as some skin diseases, such as psoriasis and acne, by means of photosensitizers that enter the body and are activated by light of appropriate wavelengths, resulting in a series of photochemical toxic effects, i.e., the generation of singlet oxygen (^1^O_2_) and other reactive oxygen species (ROS) that bind to the corresponding target tissues, leading to tissue damage and cell death. Local PDT mediated by verteporfin or indocyanine green was previously reported to reduce fat [129]. PDT acts directly and rapidly but is only applicable to superficial fat due to the limitation of light penetration. In contrast, the white fat-browning strategy has a more gradual and widespread effect. To combine these two strategies for complementary effects, the investigators obtained Pat-HBc VLP by genetically engineering the adipocyte-targeting peptide (ATP) motif (sequence CKGGRAKDC) on the surface of hepatitis B core (HBc) protein virus-like particles (VLPs), followed by a disassembly-recombination approach to loading zinc phthalocyanine tetrasulfonate (ZnPcS4) and the browning agent rosiglitazone into the particles to prepare Pat-HBc/RSG&ZnPcS4 VLPs, thus enabling simultaneous delivery of photosensitizers and browning agents to adipocytes [130]. ZnPcS4 has long-wave absorption, high quantum yield, good photostability, and a photoacoustic (PA) response, so it acts as both a photosensitizer and a PA imaging tracer in this system. Ultimately, using this complex, the investigators successfully implemented a photodynamic damage/white fat-browning strategy monitored by PA molecular imaging and fully demonstrated the strategy’s effectiveness, reliability, and safety. Although Pat-HBc is less immunogenic than wild-type HBc, it still cannot completely avoid the immune response caused by repeated administration, which needs to be addressed in the follow-up work.

Than et al. developed core–shell structured micro-spear-like polymeric drug reservoirs, named micro-lances (MLs), using a simple hot-pressing method [131]. The core was based on PLGA and NaCl loaded with rosiglitazone and CL316243 (a selective β3-adrenergic receptor agonist), and the surface layer consisted of carboxymethylcellulose and Pluronic F68. The MLs have a cylindrical shape with a length of ~4mm, a diameter of ~0.23 mm, and a tip of ~10 µm. They can be easily inserted into the white subcutaneous fatty tissue with a simple nozzle at a speed of ~1 m·s^−1^ without causing skin damage or pain or losing the drug on the skin surface. Compared to transdermal microneedle delivery, this method allows for slow release of the drug, so its treatment frequency is much lower (only twice a week). The drug has higher bioavailability and can be administered in precisely controlled doses. As demonstrated in DIO mice, MLs can effectively inhibit obesity and related metabolic disorders. Moreover, they are also expected to be used for the treatment of other diseases, such as for the delivery of anti-diabetic drugs (e.g., sulfonylureas-glimepiride), antipsychotic medications (e.g., risperidone), or anti-cancer drugs (e.g., triamcinolone for breast cancer). In short, this safe, painless, and convenient method of transdermal drug delivery may provide an unprecedented long-term at-home solution for obesity and other diseases.

In another study, the browning agent rosiglitazone and antioxidant manganese tetroxide nanoparticles (MnNPs) were integrated into poly(lactic acid)–poly(ethylene glycol) (PELA) electrospun fibers (SF@Rsg-Mn), which are 1.5 μm wide and 20 μm long [132]. After being injected into the inguinal adipose tissue of DIO mice, MnNPs at the surface of the fibers continuously scavenged adipose reactive oxygen species (ROS) and attenuated oxidative stress in adipose tissue, and rosiglitazone was continuously released for 30 days to induce adipose tissue browning for energy expenditure, thus synergistically alleviating obesity.

#### 3.3.2. Orlistat

Orlistat is a commonly used marketed anti-obesity drug. It is a gastrointestinal lipase inhibitor that inhibits the conversion of dietary fat into lipids that are available for absorption by the body, resulting in weight loss. However, preventing fat absorption can produce serious side effects, such as gastrointestinal reactions and diarrhea. As a porous colloidal material, nanostructured clay (NSC) particles have favorable anti-obesity properties, and their mechanism of action is the selective adsorption of lipid digestion products, thus impeding fat absorption in the small intestine, showing comparable activity to orlistat in DIO rats [133]. A simulated intestinal lipolysis study performed by observing changes in free fatty acid concentrations revealed a combined effect of orlistat and NSC particles, which showed a 6-fold increase in the inhibitory response compared to orlistat alone, suggesting that NSC significantly enhances the biopharmacological properties of orlistat [134].

#### 3.3.3. Thiopental Sodium

Obese patients often suffer from cardiac overload, leading to myocardial hypertrophy and ventricular hypertrophy. Myocardial hypertrophy is an essential risk factor for cardiovascular disease, in which numerous factors, such as oxidative stress and inflammation, are involved in the pathophysiological process. Thiopental sodium is a fat-soluble anesthetic that reduces lipid peroxidation or inhibits the production of reactive oxygen species by neutrophils and therefore exhibits antioxidant effects. In addition, it also possesses anti-inflammatory properties. In a study of a high-fat diet-induced cardiac hypertrophy model in obese mice, it was found that thiopental sodium lipid nanoparticles attenuated cardiac damage, hypotension, and myocardial hypertrophy induced by obesity-induced cardiac insufficiency by suppressing the inflammatory pathway and also significantly altered cardiac remodeling function, aortic function, oxidative stress, and the inflammatory response [135]. The cardioprotective effect of sodium thiopental provides evidence for future treatment of obesity and related cardiovascular complications via inflammatory pathways.

#### 3.3.4. Bindarit

WAT in obese patients releases large amounts of pro-inflammatory cytokines, including tumor necrosis factor-α (TNF-α), interleukin 1-β (IL1-β), and monocyte chemotactic protein-1 (MCP-1), leading to chronic low-grade inflammation and promoting the development and progression of obesity and its comorbidities [136,137,138]. Monocytes in peripheral blood were shown to be recruited to inflammatory adipose tissue, fatty liver, and atherosclerotic plaques [139], suggesting that monocyte-mediated drug delivery might be employable to combat obesity and its associated diseases. Our group developed a laminarin-modified bindarit nanoparticle (LApBIN) based on this strategy (Figure 4) [140]. Laminarin specifically recognizes the Dectin-1 receptor present on the surface of monocytes and macrophages. Bindarit (BIN) is a selective inhibitor of MCP-1/CCL2 and TNF-α synthesis and has been shown to be effective in the treatment of obesity, acute pancreatitis, rheumatoid arthritis, and other MCP-1/CCL2-induced inflammatory diseases. However, the poor water solubility and low oral bioavailability of BIN limit its practical application. Animal experiments confirm the successful preventive effect of the developed LApBIN oral targeted delivery system on high-fat diet-induced obesity, insulin resistance, fatty liver, and atherosclerosis with only half of the original dose of 100 mg/kg every three days. Nevertheless, further studies are needed to elucidate the transport mechanism of LApBIN from intestinal lymphoid tissue to peripheral blood mononuclear cells.

**Table 4 pharmaceutics-14-01445-t004:** Biomaterial-encapsulated synthetic drugs for anti-obesity treatment.

Drugs	Materials	Mechanism of Action	Characteristics	Main Outcomes	Reference
Rosiglitazone	PLGA nanoparticles, virus-like particles (VLPs) modified with the ATP motif, PLGA/NaCl micro-lances, and PELA electrospun fibers	Induces browning of white adipocytes and reduces inflammatory responses mediated by macrophages	Targeted drug delivery to adipocytes, reducing drug toxicity and side effects	Attenuates obesity and reduces inflammatory response	[128,130,131,132]
Orlistat	Nanostructured clay particles	Inhibits lipase activity and hinders fat absorption	Reduces toxic side effects such as gastrointestinal irritation	Attenuates obesity	[134]
Thiopental sodium	Lipid nanoparticles	Reduces inflammatory responses	Improves drug bioavailability	Ameliorates obesity-induced cardiac dysfunction and cardiac hypertrophy	[135]
Bindarit	Laminarin-modified nanoparticles	Reduces inflammatory responses	Specifically identifies monocytes and macrophages and improves drug bioavailability	Prevents obesity, insulin resistance, fatty liver, and atherosclerosis	[140]

Compared with the original small-molecule drugs, formulations with integrated functional carriers can dramatically reduce toxic side effects and improve drug bioavailability. For example, using PVA/PLGA nanoparticles to deliver rosiglitazone can solve the problem of its poor water solubility and achieve macrophage-targeted enrichment of the drug, thus reducing the inflammatory response of macrophages induced by obesity; combining orlistat with NSC particles can reduce the irritation response to the drug in the gastrointestinal tract and significantly improve the drug activity; LApBIN, which specifically recognizes monocytes and macrophages after oral administration, has demonstrated excellent therapeutic effects on a variety of inflammatory diseases, such as obesity and rheumatoid arthritis. Therefore, the functionalized drug delivery system has unique advantages in developing new dosage forms of anti-obesity drugs, and its application is expected to increase the number of weight loss drugs currently used in clinical practice, offering broad prospects for the treatment of obesity and its complications.

### 3.4. Biologic Drugs for Anti-Obesity Treatment

Numerous biological drugs, such as peptides, miRNAs, cytokines, etc., have the functions of inhibiting the development of inflammation, controlling lipid metabolism, and regulating glucose uptake (Table 5). For example, interleukins IL-4 and IL-10 induce an increase in M2 macrophages and a decrease in the secretion of pro-inflammatory cytokines; the pro-apoptotic peptide KLA disrupts cellular mitochondrial function, resulting in the release of cytochrome C and the induction of apoptosis. PDBSN, a bioactive peptide, can inhibit adipocyte differentiation, and miRNA drugs such as miR33 and miR-130b can regulate lipid metabolism. However, these biological drugs all face stability and targeting problems in vivo. To address this problem, researchers have coupled small-molecule drugs to polymers or applied carrier materials such as liposomes and polymeric scaffolds to prolong the half-life of drugs, alter drug release profiles, absorption, and distribution, etc., to improve the drug utilization rates and safety.

#### 3.4.1. Targeting Adipose Tissue Macrophages

Adipose tissue macrophages (ATM) play a vital role in developing obesity-induced chronic inflammation. The proportion of macrophages infiltrating adipose tissue in obese populations is substantially higher than in normal individuals. These macrophages surround adipocytes in a coronal structure. They are predominantly M1 type and release large amounts of pro-inflammatory factors that cause obesity-related chronic diseases such as T2D and atherosclerosis [141,142].

Interleukin-4 (IL-4) has been shown to induce an increase in M2-type macrophages, as well as Th2 and regulatory T cells [143,144]. Porous poly(lactide-co-glycolide) (PLG) implants coated with human interleukin-4 (hIL-4)-expressing lentivirus were transplanted into epididymal WAT of HFD-induced early obese mice [145]. It was found that locally expressed HIL-4 would induce an anti-inflammatory phenotype in adipose tissue macrophages, and an increased proportion of helper T cells was observed in adipose tissue, revealing its potential in regulating adipose tissue inflammation and metabolism. The anti-inflammatory cytokine interleukin-10 (IL-10) reduces the secretion of pro-inflammatory cytokines in macrophages through a STAT3-dependent pathway. It has therapeutic effects on diabetes and various inflammations, such as psoriasis and inflammatory bowel disease [146,147]. Delivering it to macrophages in adipose tissue is expected to alleviate obesity-related chronic inflammation. However, the short half-life of IL-10 makes this a challenge. Phosphatidylserine (PS) is an “eat me” signal expressed on the surface of apoptotic cells and can be recognized by various phagocytosis receptors on the surface of macrophages. R. Toita et al. prepared liposomes with PS modified on the surface (PSL) to deliver IL-10 to macrophages, and IL-10-conjugated PSL (PSL-IL10) had a high affinity for macrophages [148]. Experiments on HFD-induced obese mice showed that PSL-IL10 significantly reduced total serum cholesterol, altered adipocyte size, and inhibited the secretion of pro-inflammatory cytokines such as IL-6 and TNF-α in adipose tissue compared with IL-10 and PSL alone, revealing the good efficacy of the synergistic effect of IL-10 and PSL on obesity.

Sea cucumber saponins, secondary metabolites of sea cucumber, have been shown to alleviate obesity and hepatic steatosis and restore glucose tolerance in DIO mice [149]. They are highly water-soluble, which reduces their permeability and absorption in the intestine and largely limits their bioavailability. The use of liposomes can overcome this drawback. Studies have shown that sea cucumber saponins loaded into liposomes exhibit better performance than the common form of sea cucumber saponins in terms of weight loss, hypolipidemia, and alleviation of insulin resistance. In addition, sea cucumber saponin liposomes can reduce the release of pro-inflammatory cytokines and macrophage infiltration in obese mice, thus effectively reducing adipose tissue inflammation, which is not possible with their common form [150].

#### 3.4.2. Inhibition of WAT Angiogenesis

Hypertrophy and hyperplasia of adipocytes in WAT is the main feature of obesity development, and this process requires neovascularization to provide adequate nutrients and oxygen. Therefore, obesity and its related diseases can be treated by inhibiting angiogenesis in WAT [151]. The antiproliferative protein prohibitin (PHB) is overexpressed in the vascular system of obese mice as well as in human WAT. It can be used as a target for the delivery of cytotoxic drugs to vascular endothelial cells to inhibit angiogenesis and reverse obesity [152]. CKGGRAKFC peptide, also known as adipose homing peptide (AHP), exhibits high specificity for PHB. Injection of AHP-coupled pro-apoptotic peptide KLA (AHP-KLA) into DIO mice and monkeys resulted in significant weight loss and the reversal of obesity [153,154]. After AHP-KLA enters cells through receptor-mediated endocytosis, it disrupts mitochondrial function and leads to the release of cytochrome c, which in turn activates the caspase-3 pathway to induce apoptosis. However, chimeric peptides are unstable in vivo and require higher doses to maintain efficacy. Moreover, long-term systemic administration will cause the body to produce antibodies against chimeric peptides, resulting in lower efficacy in later stages. The application of nanocarriers can increase the stability of peptide drugs. For example, liposomes composed of phosphatidylcholine and cholesterol were loaded with KLA, and the surface of the liposomes was modified with AHP. After 18 days of treatment, PHB-mediated endothelial cell uptake of the liposomes resulted in a significant reduction in body weight of 14% in DIO mice [155], compared to 5% when using AHP-KLA itself [153]. The decline in body weight in mice was accompanied by decreased leptin levels, reduced macrophage numbers, and reduced angiogenic clusters. None of these changes were evident in mice treated with AHP-KLA during the same period.

#### 3.4.3. Regulation of Signaling Pathways

The AMPK (AMP-activated protein kinase) pathway is a signaling pathway that coordinates adipogenic differentiation and plays an important role in maintaining metabolic homeostasis [156]. PDBSN (sequence GLSVADLAESIMKNL) is a bioactive peptide found to inhibit adipocyte differentiation by activating the AMPK pathway. The peptide was encapsulated in liposomes with surface-modified visceral tissue-targeting peptide and cell-penetrating peptide for in vivo delivery, thus improving its circulation stability and specificity to adipose tissue [157]. Experiments on DIO mice showed that the treatment significantly reduced adipose tissue mass, especially visceral adipose tissue. In addition, glucose metabolism and dyslipidemia were modified.

miR33, a microRNA, controls lipid metabolism by regulating the expression of leptin, insulin, and lipoproteins. To transfer miR33 to efficiently target cells for successful gene therapy, a degradable, biocompatible block copolymer poly (citric acid)-glycerol-polylysine (PCG-EPL) was synthesized (Figure 5) [158]. In vitro cellular experiments demonstrated that the self-assembled PCG/miR33 anti-obesity nanocomplex formed could effectively deliver miR33 to adipocytes and reduce the expression of IL-1β, which is associated with obesity. Therapeutic experiments on DIO rats showed that the nanocomplex reduced the expression of inflammatory factors such as IL-1β, TNF-α, and IL-6 and enhanced lipid metabolism in rats without suppressing their appetite, effectively reducing body weight. This study certainly brings good news for obese patients who cannot control their weight through an autonomous diet. miR-130b also exerts a regulatory effect on lipid metabolism. After intravenous injection of microvesicle-packaged miR-130b (miR-130b-MV) into a DIO C57BL/6 mouse model, miR-130b was detected to be delivered to epididymal adipose tissue, where it downregulated PPARγ protein content while upregulating lipolytic genes, hormone-sensitive lipase, monoglyceride lipase, and leptin [159]. This resulted in a significant reduction in fat deposition and the partial restoration of glucose tolerance. Further studies are needed to assess the cytotoxicity and half-life of miR-130b-MV in blood to facilitate the development of corresponding drugs.

#### 3.4.4. Regulation of Hormone Levels

As mentioned, leptin is a hormone secreted by adipocytes and acts upon the brain’s hypothalamus to regulate appetite and energy metabolism. The progression of obesity leads to a decrease in the transport of leptin across the blood–brain barrier (BBB). To improve the binding efficiency to its receptor, leptin with N-terminal amine modification of Pluronic P85 (LepNP85) was prepared [160]. Leptin with N-terminal amine modification of PEG (LepNPEG5K) was synthesized for use as a control. Intranasal administration (INB) was adopted to bypass the BBB. After dosing, LepNP85 conjugates accumulated significantly more in the hypothalamus and hippocampus of the brain than natural leptin and LepNPEG5K, suggesting that Pluronic P85 modification combined with INB administration improved brain delivery of leptin. In addition, a non-covalent mixed intranasal formulation of leptin and cell-penetrating peptides (CPPs) was also developed [161]. An amphiphilic CPP was used here, namely, L-permeabilin (sequence RQIKIWFQNRRMKWKK). The non-covalent mixing method was used with consideration of the difference in the mode of cellular internalization of the two, where an energy-independent pathway mainly mediates the cellular internalization of peptide drugs, and CPPs can enter the cells through endocytosis (e.g., micropinocytotic action). Animal experiments have shown that leptin in combination with L-permeabilin can effectively accumulate in the anterior part of the brain, suppress appetite and weight gain in rats, and reduce plasma triglyceride levels. These are achieved by stimulating Stat3 phosphorylation of leptin receptors [162].

Oxyntomodulin (OXM), a 37-amino-acid peptide enteroendocrine hormone, is a dual agonist of the glucagon receptor (GCGR) and glucagon-like peptide 1 (GLP-1) receptor (GLP-1R) with effects on the regulation of glucose metabolism, insulin secretion, food intake, and energy expenditure [163]. Natural OXM is rapidly degraded in vivo within minutes and quickly cleared by the kidneys, limiting its further clinical application. A series of new OXM analogs were prepared by modifying the structure of the middle and C-terminus of the peptide sequence to enhance the affinity for the receptor, and these analogs were further chemically coupled with PEG to prolong their half-life for sustained release in vivo [164]. Pharmacological studies of these PEGylated analogs in a dietary obesity mouse model showed that analog 10 had the most potent hypoglycemic and weight-lowering efficacy, normalizing lipid metabolism and hepatic steatosis. Thus, they are up-and-coming candidates for anti-T2D and anti-obesity treatments.

#### 3.4.5. Genetically Engineered Drug Delivery

Islet amyloid polypeptide (IAPP) and irisin are potential browning hormones for the treatment of obesity. IAPP, also known as pancreatic precipitin, is released from the pancreas, increasing energy expenditure [165]. Iridin effectively converts WAT to brown adipose tissue and improves glucose tolerance, triggering a thermogenic program [166]. However, protein-based drugs have a short half-life, require frequent administration, and might even cause harmful immune responses. Therefore, plasmids containing both IAPP and iridoid gene structures were developed. Linear polyethyleneimine was used as a gene delivery vehicle, which is considered the gold standard for polymer-based gene delivery with low toxicity, easy modification, low immunogenicity, and lysosomal escape. It was confirmed that combinatorial gene therapy had a synergistic effect on weight loss in DIO mice [167]. Delivery of this gene therapy system by intraperitoneal injection enhanced the anti-obesity thermogenic program, increased energy expenditure, and enhanced the expression of the browning genes Ppargc1a, Prdm16, and Pparg. This cationic polymer-based dual-browning gene vector exemplifies the advantages of combination therapy against obesity, a complex disease caused by multiple factors.

Fibroblast growth factor 21 (FGF21) is an endocrine hormone produced mainly by the liver and adipose tissue with the function of regulating glucose metabolism, lipid metabolism, and insulin resistance. Natural FGF21 has therapeutic potential for obesity and diabetes. Its plasma half-life is quite short, ranging from 0.5 to 5 h, depending on the administration route and species, which is a great challenge as a therapeutic protein [168]. Coupling protein drugs to PEG is an effective strategy to extend their half-life, but this raises the cost of PEG derivatization and the in vitro covalent coupling and purification steps. Moreover, PEG accumulation in vivo can cause side effects such as renal epithelial vacuolization. The polypeptide fusion technique was used to combine a polypeptide (PsTag) containing a repeatable sequence of five amino acids (Pro, Ser, Thr, Ala, and Gly) with FGF21 for fusion expression in E. coli. PsTag has physicochemical properties resembling PEG, i.e., uncharged, hydrophilic, and flexible. In addition, it has no potential cumulative toxicity. Studies on DIO mice found that the PsTag-FGF21 fusion protein had an extended half-life of 12.9 h and produced significantly better weight loss and hypoglycemic effects than natural FGF21 while reversing hepatic steatosis [169].

Silencing fatty acid-binding protein 4 (fabp4) ameliorates metabolic abnormalities and results in weight loss in DIO mice [170]. To selectively silence fabp4 in white adipocytes, a CRISPR interference (CRISPRi) system (dCas9/sgFabp4) modified with a fusion peptide composed of ATP motif and 9-polyarginine (ATS-9R) was developed [171]. After intraperitoneal injection, the accumulation of (dCas9/sgFabp4)-(ATS-9R) in adipose tissue and CRISPRi system-mediated Fabp4 silencing were observed in obese mice, resulting in reduced inflammation in adipose tissue, weight loss, and the restoration of hepatic steatosis. This reveals that directing the CRISPRi system of catalytically dead Cas9 and Fabp4 sgRNAs to adipose tissue by using adipose tissue-targeting peptides is a potentially effective strategy to treat obesity and obesity-induced metabolic syndrome (Figure 6).

Biological drugs have the characteristics of high pharmacological activity and minor side effects. However, their use is limited by their poor stability, tendency to deactivate in acid–base environments or by the action of enzymes in vivo, or their inability to cross the BBB. Combining these drugs with targeted biomaterials can improve the stability of drugs and reduce side effects on normal tissues. For example, hIL-4-expressing lentivirus was loaded onto a PLG scaffold for sustained release to modulate adipose tissue inflammation; IL-10, a small anti-inflammatory molecule, was loaded into liposomes modified with phosphatidylserine (PS), specifically recognized by macrophages to exert anti-inflammatory functions; leptin uptake in the brain was increased by coupling a membrane-penetrating peptide with it and using an INS delivery strategy to bypass the BBB; a PsTag-FGF21 fusion protein was prepared using a polypeptide fusion technique to significantly increase the half-life of FGF21. These studies provide effective and diverse strategies for developing more clinically available anti-obesity biologics, which are expected to alleviate the distress of obese patients.

**Table 5 pharmaceutics-14-01445-t005:** Biologic drugs for anti-obesity treatment.

Strategies	Drugs	Materials	Mechanism of Action	Characteristics	Main Outcomes	Reference
Targeting adipose tissue macrophages	HIL-4	PLG implants	Induces an anti-inflammatory phenotype in macrophages, increase the proportion of helper T cells	Acts locally and long-lastingly on WAT	Attenuates obesity and reduces inflammatory response	[145]
	IL-10	Liposomes with surface modified by phosphatidylserine	Inhibits the secretion of pro-inflammatory factors such as IL-6 and TNF-α	High affinity for macrophages	Attenuates obesity and reduces inflammatory response	[148]
	Sea cucumber saponins	Liposomes	Inhibits secretion of pro-inflammatory cytokines, reduces macrophage infiltration, and increases glucose uptake	Improves drug bioavailability	Attenuates obesity, reduces inflammatory response, and alleviates glucose intolerance	[150]
Inhibition of WAT angiogenesis	Pro-apoptotic peptide KLA	Adipose homing peptide; liposomes modified by fat homing peptide	Inhibits angiogenesis	Targeting the WAT vascular system	Attenuates obesity	[153,154,155]
Regulation of signaling pathways	Bioactive peptide PDBSN	Liposomes modified with visceral tissue-targeting peptide	Activates AMPK pathway to inhibit adipocyte differentiation	Targeting white adipocytes	Attenuates obesity and modulates glucose and lipid metabolism	[157]
Regulation of signaling pathways	miR33	PCG-EPL micelles	Regulates leptin, insulin, and lipoprotein expression and controls metabolism	Delivers miRNA to adipocytes	Attenuates obesity and alleviates glucose intolerance	[158]
	miR-130b	Cellular microvesicles	Regulates lipid metabolism	Delivers miRNA to epididymal adipose tissue	Attenuates obesity, alleviates glucose intolerance	[159]
Regulation of hormone levels	Leptin	Pluronic P85	Acts on the hypothalamus to regulate appetite and energy metabolism	Polymer modification to extend the half-life of the drug	Attenuates obesity	[160]
	Oxyntomodulin analog	PEG	Regulates glucose metabolism and insulin secretion	Polymer modification to extend the half-life of the drug	Attenuates obesity, reduces blood glucose level, and reverses liver steatosis	[164]
Genetically engineered drug delivery	Plasmids containing both IAPP and iridoid gene structure	Linear polyethyleneimine	Induces browning of white adipocytes and triggers thermogenic procedures	Synergistic effects of combination gene therapy	Attenuates obesity	[167]
	FGF21	PsTag polypeptide	Regulates glucose metabolism, lipid metabolism, and insulin resistance	Preparation of E. coli fusion expression proteins of cytokines and polypeptides to extend the half-life of protein drugs	Attenuates obesity, reduces blood glucose level, and reverses liver steatosis	[169]
	fabp4	dCas9/sgFabp4 CRISPRi interference system	Silencing the gene of Fabp4	CRISPRi interference technology for selective regulation of metabolism-related genes	Attenuates obesity and reverses liver steatosis	[171]

## 4. Conclusions

Over the past decades, although researchers have made great efforts, conventional obesity treatments are still inadequate to maintain metabolic homeostasis and prevent life-threatening complications, so there is an urgent need for therapies with higher efficacy and specificity. The continuous development of drug delivery systems is essential for advances in agent-based disease treatment, and various types of carrier materials, such as liposomes, micelles, or vesicles, and transdermal microneedles have been developed. Combining anti-obesity drugs with these materials can improve drug stability, prolong their half-life, increase drug enrichment in specific cells or tissues, and reduce adverse effects. While these biomaterial-engineered targeted agents have achieved good efficacy in mouse models of high-fat diets, clinical transformation is still a challenge. Further comprehensive assessments are required to determine whether the use of these materials in humans raises safety issues and whether they have good efficacy in humans and require long-term administration to achieve stable weight loss. This means that specific criteria are needed to assess the safety and effectiveness of these dosage forms in humans. Finally, it should be emphasized that the selection of appropriate animal models in therapeutic studies of obesity and its comorbidities is highly influential. There are many types of obesity models available, including genetic animal models (monogenic, polygenic, and transgenic obesity models) and non-genetic animal models (diet-induced, exotic, large animals, and surgical obesity models). They each have their own advantages but are also accompanied by limitations, so the study should be selected according to the specific type of obesity and the etiologic/pathologic mechanism of the particular comorbidity.

## Figures and Tables

**Figure 1 pharmaceutics-14-01445-f001:**
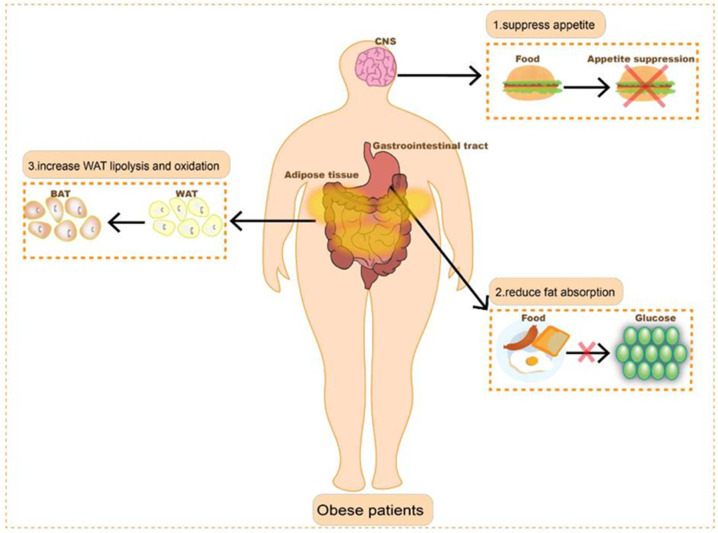
Main mechanisms of action of anti-obesity drugs.

**Figure 2 pharmaceutics-14-01445-f002:**
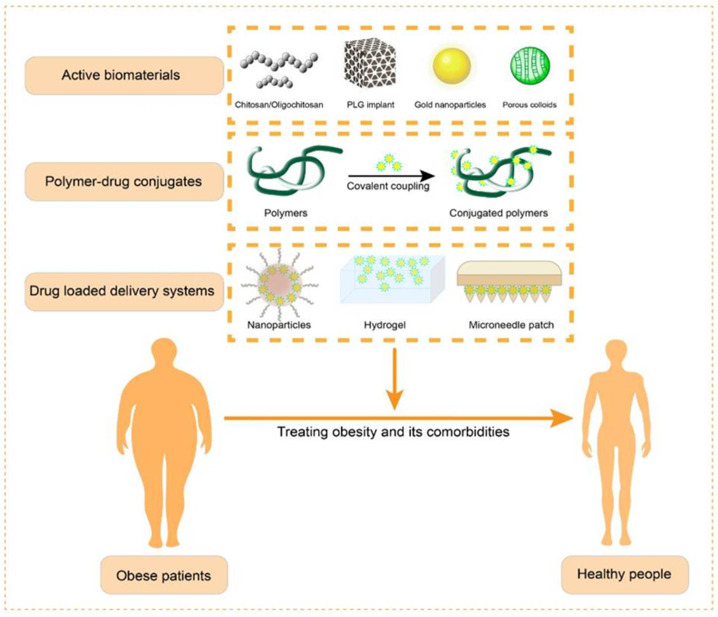
Main strategies for the application of biomaterials to combat obesity and its comorbidities.

**Figure 3 pharmaceutics-14-01445-f003:**
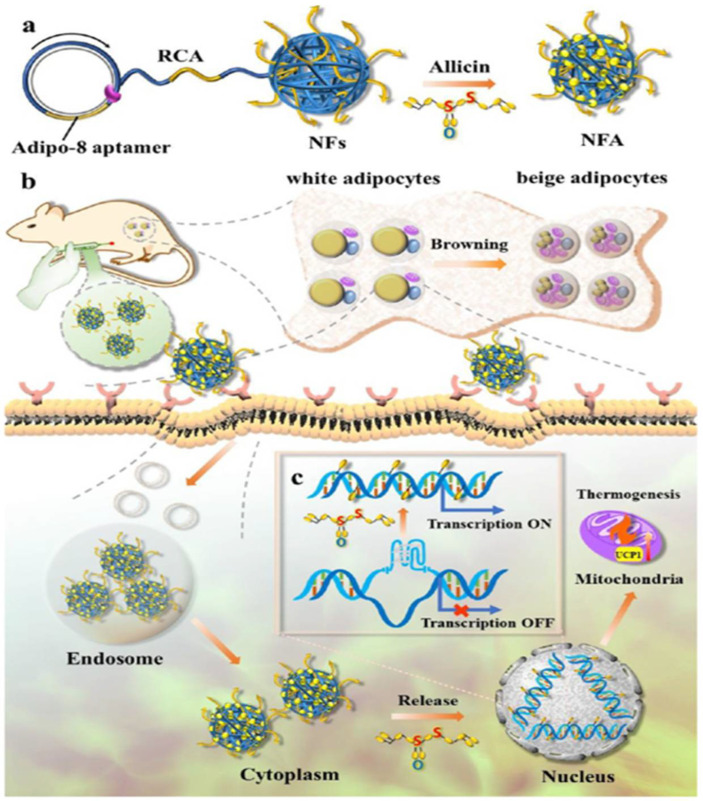
Schematic illustration of effective obesity restraint through targeted activation of G4-mediated UCP1 expression. (**a**) Schematic illustration of the synthesis process of aptamer-functionalized binary-drug delivery system. (**b**) The NFA binds to the target receptor (adipocyte plasma membrane-associated protein) in the cellular membrane. (**c**) Allicin decreases the stability of G454 in the UCP1 promoter, increasing UCP1 expression in adipose cells. Upon entering the cytosol, intracellular target (G454) recognition events trigger the expression of UCP1 of adipose cells, increasing systematic energy expenditure. Reproduced with permission [126]. Copyright 2022, American Chemical Society.

**Figure 4 pharmaceutics-14-01445-f004:**
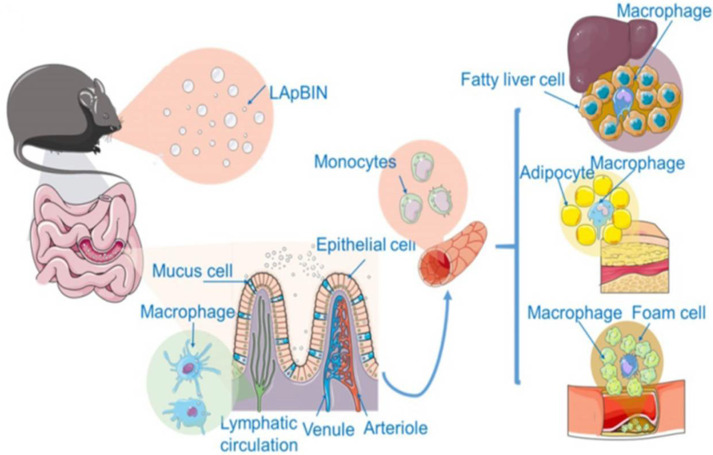
Schematic diagram of LAM-mediated oral targeting of nanoparticles to diseased sites of obesity-related diseases distant from the gastrointestinal tract. Reproduced with permission [140]. Copyright 2020, Ivyspring International.

**Figure 5 pharmaceutics-14-01445-f005:**
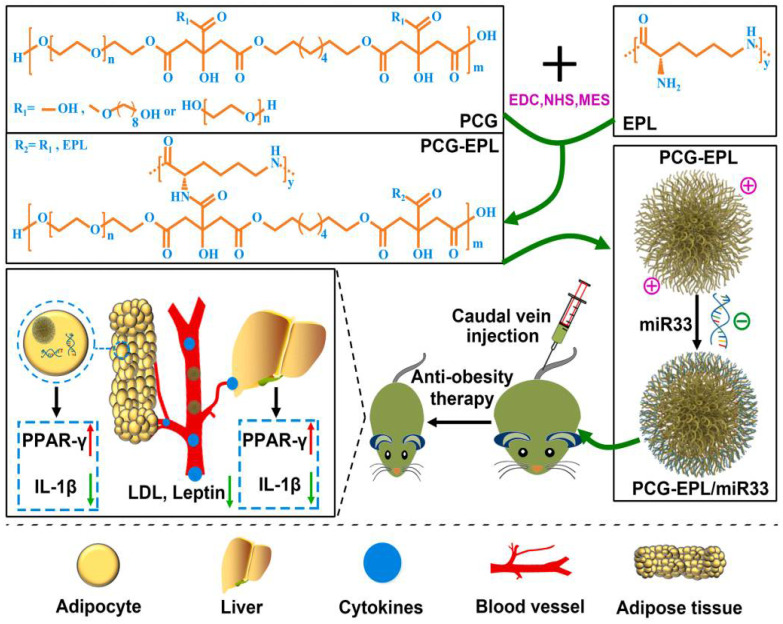
Schematic depiction for the synthesis of PCG-EPL/miRNA and the application of PCG-EPL/miR33 agonist in obesity therapy. Reproduced with permission [158]. Copyright 2021, Elsevier.

**Figure 6 pharmaceutics-14-01445-f006:**
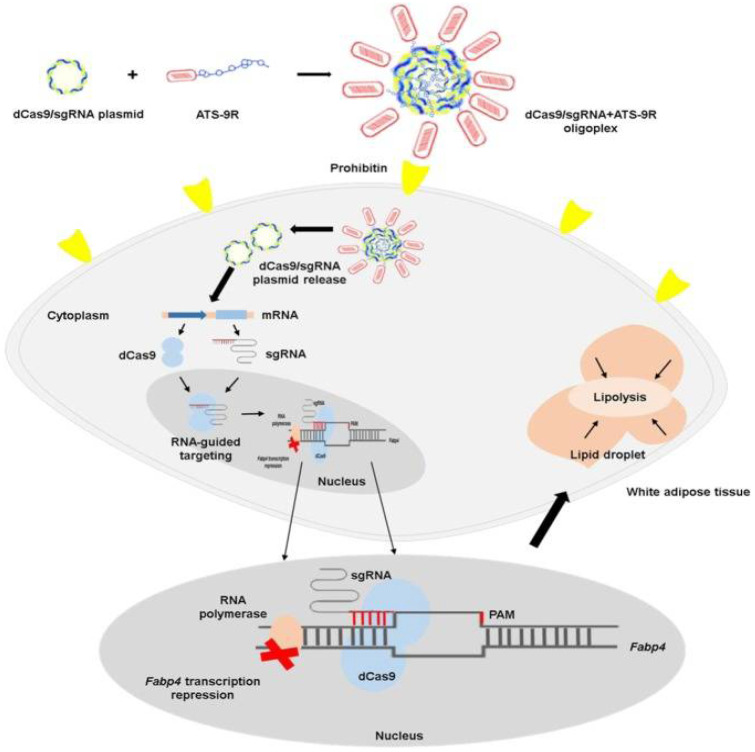
Schematic illustration representing nonviral CRISPR interference system delivery to white adipocytes. The dCas9 plasmid and sgRNA against the Fabp4 gene formed complexes with ATS-9R peptide via electrostatic interaction. The highly cationic ATS-9R functions as a targeting and condensing peptide to deliver dCas9/sgRNA to white adipose tissues, and dCas9 plasmid contains nuclear localization signals (NLSs) on both the N-terminus and C-terminus for delivery to the nucleus. Reproduced with permission [171]. Copyright 2019, Cold Spring Harbor Laboratory Press.

**Table 1 pharmaceutics-14-01445-t001:** Anti-obesity drugs approved by US FDA.

Drug	Mechanism of Action	Delivery Mode	Side Effects	Reference
Orlistat	Increase intestinal lipid excretion and block the absorption of fat	Oral administration	Diarrhea, flatulence	[34]
Phentermine	Work through central nervous system pathways to reduce appetite		Insomnia, constipation, palpitation, dry mouth	[35,36]
Lorcaserin			Headache, dizziness, fatigue, nausea, dry mouth	[37]
Naltrexone/bupropion sustained-release			Nausea, headache, constipation, dizziness, vomiting, dry mouth	[38,39]
Phentermine/topiramate extended release			Insomnia, constipation, dizziness, taste disorders	[40]
Liraglutide	Induce satiety by delaying gastric emptying	Hypodermic injection	Nausea, vomiting, diarrhea, constipation, dyspepsia	[43]
Semaglutide			Nausea, bloating, diarrhea, and vomiting	[45,46]

## Data Availability

This study did not report any data.

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
