# Peer review of "Biomaterial-Based Therapeutic Strategies for Obesity and Its Comorbidities"

_pharmaceutics, 2022, doi:10.3390/pharmaceutics14071445_

Round 1

Reviewer 1 Report

This is a well-organized and well-illustrated review paper, has an important clinical message. The review focused on the limitations of currently available obesity drugs and the pre-clinical and clinical research progress of anti-obesity drugs. Paragraphing is concise and good, and the article consists of major recent advancements in the field of obesity drug research and deserves publication after some revisions listed below.

1.    Lines 15-17. I suggest the authors to revise these lines in the abstract, as not all anti-diabetic medications can cause serious mental and/or cardiovascular complications. Also, there are several well tolerated drugs with a good clinical outcome with a high efficacy like some GLP-1R agonist ex: Semaglutide.

2.    Lines 54-59- Please add proper reference

3.    Lines 63-65- Please add reference

4.    For section 2. Pharmacological treatment of obesity, I suggest the authors to tabulate the list of approved drugs, mechanism of action and their side effects or complications with proper references so that it is easy for the readers to understand.

5.    I would suggest the authors to briefly describe about the importance of choosing the animal model strains in diabetic research as some research models do not transform for application of antidiabetic drugs in humans that have a good efficacy in preclinical models

Author Response

C: Thanks so much for your careful review with helpful suggestions for improving our work. We have revised our manuscript and all changes were marked out with Red fond. A point-by-point response was listed below to answer the issues clearly.

Q1. Lines 15-17. I suggest the authors to revise these lines in the abstract, as not all anti-diabetic medications can cause serious mental and/or cardiovascular complications. Also, there are several well tolerated drugs with a good clinical outcome with a high efficacy like some GLP-1R agonist ex: Semaglutide.

A: As suggested, we have changed the sentence “Current anti-obesity therapeutics are mostly focusing on reducing food intake by suppressing the appetite, which leads to a limited efficacy for obesity-related diseases, but serious mental and/or cardiovascular side effects, thus alternative treatment strategies are in urgent need.” into “Pharmacotherapy alone or combined either with lifestyle alteration or surgery, represents the main modality to combat obesity and its complications. However, most anti-obesity drugs are limited by their bioavailability, target specificity and potential toxic effects. Only a handful of drugs, including orlistat, liraglutide and semaglutide, are currently approved for clinical obesity treatment. Thus, there is an urgent need for alternative treatment strategies.”

Q2. Lines 54-59- Please add proper reference

A: We had added the proper references, those are references [14-17], and marked them in RED fond in the revised version of manuscript.

Q3. Lines 63-65- Please add reference

A: We had added the proper references, those are references [18,19], and marked them in RED fond, too.

Q4.  For section 2. Pharmacological treatment of obesity, I suggest the authors to tabulate the list of approved drugs, mechanism of action and their side effects or complications with proper references so that it is easy for the readers to understand.

A: As suggested, the corresponding content was added, which are shown in Table 1. 

Q5. I would suggest the authors to briefly describe about the importance of choosing the animal model strains in diabetic research as some research models do not transform for application of antidiabetic drugs in humans that have a good efficacy in preclinical models.

A: Thank you for your suggestion, we have discussed this in the section of conclusion. We added the sentence “Finally, it should be emphasized that the selection of appropriate animal models in therapeutic studies of obesity and its co-morbidities is highly influential. There are many types of obesity models available, including genetic animal models (monogenic, polygenic and transgenic obesity models) and non-genetic animal models (diet-induced, exotic, large animals and surgical obesity models). They each have their own advantages but are also accompanied by limitations, so in the study should be selected according to the specific type of obesity and the etiologic-pathologic mechanism of the certain co-morbidity.”

Reviewer 2 Report

The submitted manuscript is a comprehensive and well-written review with a detailed overview of the literature on biomaterial-based therapeutic strategies for obesity and its comorbidities.

I enjoyed reading the manuscript. I commend the authors for several strengths of their work, including addressing an exciting and timely question. The subject is in the range of the journal, and the manuscript is of clinical relevance.

I have only one minor suggestion.

I recommend dedicating a bit more space to the definition and diagnosis of obesity. Obesity is usually defined as an excessive or abnormal accumulation of body fat that adversely affects health. The adipose tissue dysfunction in obesity leads to low-grade chronic inflammation. The risk does not lie so much in the amount of fat accumulated as in its distribution. Visceral obesity is an important risk factor for cardiovascular diseases, metabolic disorders, and cancer

Author Response

Thanks for your helpful suggestions. We have modified our discourse on the definition of obesity, accordingly.